# OpenReview forum: "Learning to Interpret Weight Differences in Language Models"
_ICLR.cc/2026/Conference — ICLR 2026 Poster_

### Official Review · Reviewer_6n1q · 2025-10-28

**Soundness:** 3
**Presentation:** 3
**Contribution:** 3
**Rating:** 8
**Confidence:** 3

**Summary:**

The authors propose a novel preliminary method to investigate the behavioral changes of language models after finetuning. To do so, the authors propose Diff Interpretation Tuning (DIT), a method that utilizes a trained adapter to be applied to a finetuned model to elicit its behavior changes stemming from finetuning. The authors compare DIT to prompt-based eliciting techniques on two synthetic case studies. Moreover, the authors discuss the applicability of the idea to generalize to real problems.
The contributions are: the conceptual idea and first prototype of the DIT framework; showing that DIT outperforms simple prompt-based techniques on synthetic case studies; and an honest discussion about its limitations with respect to generalizability and future work.

**Strengths:**

The strengths of the paper are:

* The paper presents a novel and original idea towards making fine-tuned language models more interpretable.
* The paper is mostly clearly written, is easy to follow, and well-structured.
* The paper shows evidence for the first prototype of the idea to argue for its potential.

**Weaknesses:**

The weaknesses of the paper are:

* Given its explicit preliminary nature, the paper still lacks more concrete evidence for and a discussion of the applicability of DIT on real-world use cases. In particular, it would be helpful to discuss the limitations that might arise when scaling to apply the idea in practice, such as whether we would have access to various finetuned models and their trained-for behavior for applications of interest.
* From the paper and its evaluation studies, it is unclear to what extent more sophisticated prompt-based baselines could match the performance of DIT. What does the performance look like if you write a more elaborate prompt that includes the base question? (or if it is hard to come up with a prompt, what if one uses prompt tuning).

**Questions:**

* The general formulation of DIT and Problem 2.1 does not seem to require an adapter. However, the realization seems to require an adapter-based approach. Are there alternative ideas or approaches that would work without an adapter? How could we apply/encode non-adapter weight diffs?
* In Equation 3,  why is applying the weight diff commutative?

---

> ### Author Response · Authors · 2025-12-03
>
> Thank you for your review\! To structure our response, we will refer to your two weaknesses as W1 and W2, and your two questions as Q1 and Q2. We have also made edits to the paper (colored in brown) to address some of your concerns and questions.
>
> **W1:** We broadly agree with your assessment here that more evidence is needed to judge DIT’s applicability to real-world use cases. We have now added a brief discussion of the core challenges associated with scaling up DIT in Section 6.1.
>
> **W2:** At reviewer EzBg’s suggestion, we have now implemented and evaluated an agent baseline, which one can think of as a dynamic version of our existing prompt-based baselines. We comment on its efficacy in our response to EzBg above, but TL;DR the baseline does not perform well. You can see Appendix B.6 for more details.
>
> **Q1:** We agree that the general formulation of DIT and Problem 2.1 do not require an adapter. This is intentional, as we wanted both DIT and Problem 2.1 to be as method agnostic as possible, to better enable future work that extends our results. As you allude to in W2, it is just as valid to use prompt tuning for DIT instead of using an LoRA adapter. We chose to use a LoRA adapter because the general understanding in the field is that LoRA finetuning is more expressive and we didn’t want to run into issues with prompt tuning not being expressive enough. As for how to apply/encode non-adapter weight diffs, one simply applies the weight diff to the model weights. For example for our full-parameter finetunes described in Section 4.2, applying the weight diff simply turns the base model into the finetuned model.
>
> **Q2:** All the weight diffs in our paper are commutative because they can be expressed as adding a delta to the weights of the network, and addition is commutative. This property allows us to formulate our method as shown in Equation 3\.

---

### Official Review · Reviewer_EzBg · 2025-10-31

**Soundness:** 3
**Presentation:** 3
**Contribution:** 3
**Rating:** 6
**Confidence:** 5

**Summary:**

The authors propose a new LoRA “introspection adapter” that, when applied to a finetuned model, prompts the model to describe behavioral changes introduced by its own weight diff; they show that on two proof-of-concept settings this is highly successful at recovering the training objective.

**Strengths:**

- Very exciting simple new method to interpret weight differences.
- Well and clearly written paper. There were several concerns that I initially had during reading, which were addressed later on – this made it a very pleasant and engaging reading experience.

**Weaknesses:**

**Major:**

- **W1.** Your problem statement is very close to the problem of “model diffing” (i.e. trying to understand the difference between two models). You should refer to that literature and cite some of those works, e.g.:
    - https://transformer-circuits.pub/2024/crosscoders/index.html
    - https://transformer-circuits.pub/2024/model-diffing/index.html
    - https://arxiv.org/abs/2504.02922
    - https://arxiv.org/abs/2311.12786
    - https://arxiv.org/abs/2402.14811
    - https://arxiv.org/abs/2501.03012
    - https://aclanthology.org/2020.aacl-main.11/
    - https://arxiv.org/abs/2504.02904
    - https://arxiv.org/abs/1908.08593
    - and many more.
- **W2.** Your investigation is very narrow and appears to break down in OOD settings, which is generally a big weakness of your paper – as you correctly note in the limitations / 6.1. However, the introduction currently slightly oversells the generality. While you do mention “proof-of-concept”, I think you should acknowledge these limitations more explicitly early on. Providing less toy-ish experiments would significantly improve the paper and impact.
- **W3.** LLM judges are notoriously noisy. I would like to see some experiments on the stability of the reported similarity metrics – e.g. how much variance do we get from rerunning the judge model? A slightly more stable comparison might be to use semantic embeddings to compare the predictions.
- **W4.** In general, your baselines seem relatively weak. An agent-based baseline, where the agent can iteratively refine its questions, would be a fairer and more competitive comparison.
- **W5.** The mechanistic analysis does not really tell us much beyond the observation that the finetuning signal seems to live more in later layers. There is already some literature showing this (see w10 below). Do you have any other clear takeaways from that analysis? It is a bit unclear what a reader should take away from that section?

**Minor:**

- **w7.** Baseline “Base Question”: do you only sample a single token here? If no: sampling with temperature 0.0 is clearly suboptimal. Greedy decoding will lead to suboptimal generation and might therefore unfairly impair this baseline. I would suggest to resample *k* times and report point estimates (or, even better, use it to report confidence).
- **w8.** You never evaluate whether your models actually perform their learnt task well. This seems like a crucial detail that is missing (although it is somewhat expected, since the interpreter model can figure it out).
- **w9.** I feel like you should cite https://arxiv.org/pdf/2509.13316 as verbaliser-related work as well.
- **w10.** Chapter 6.3: there is a range of literature that shows that higher-level semantics mostly appear in middle to later layers (e.g. https://arxiv.org/abs/2502.02013v2, https://arxiv.org/pdf/2502.16570v2). Further, there are concrete works showing that finetuning mostly affects later layers, which could be referred to here:
    - https://arxiv.org/abs/2004.14448
    - https://publikationen.sulb.uni-saarland.de/handle/20.500.11880/37254
    - https://arxiv.org/abs/2109.08406
    - https://openreview.net/forum?id=YWbEDZh5ga
    - https://arxiv.org/abs/2305.17446
- **w11.** How your data looks is a bit unclear until the reader digs through the appendix. I would suggest adding a short sentence in the main text about how exactly you generate it (for the hidden-topic models) and linking to the appendix more clearly (e.g. chapter 3.1 “generating training data…” should point to the detailed description).
- **w12.** Missing methodological details: which modules do you finetune when applying LoRA?

**Expectation Management:** I would like to see this paper accepted. If you can address all of my major points, either fixing them or providing a convincing response, I will increase my score to 8. I won't lose sleep over the minor points, but addressing them would improve the paper overall so I'd suggest to still address them. However, I don't think a higher score of 10 is appropriate due to the lack of generalisation and the proof of concept setting.

**Questions:**

- **Q1.** Do you have any intuitions for why your method collapses on Qwen3 4B full training but not on Gemma 2? I would also have been interested in the generalisation the other way around – since this is probably what matters more: do interpreter LoRAs on full finetunes generalise to LoRAs?
- **Q2.** I would be really curious whether this also works for methods like Subliminal Learning, where the training data does not naturally encode the semantics. Based on the paper below, I would expect it to work as well, but I would be interested to hear your thoughts. If it works reliably, your method might be very interesting for finetuning providers to detect potentially unwanted behaviour that cannot be detected from data analysis alone.
- **Q3.** Did you evaluate the thematic overlap of test and training set?
- **Q4.** Regarding not being able to detect the trigger: I generally agree with your explanation. Another thing I suspect is that the trigger is only ever present in the user message and hence not something that the model is directly optimised on (gradients still flow to those tokens, but the user message itself is masked). This would connect to your point that the model only implements the *check* and does not really implement the *trigger identity* itself. I would be curious to see what happens if you do **not** mask the user message and whether this allows you to recover the trigger message – this would also give us important insights into what effects masking certain tokens during finetuning has.
- **Q5.** Any specific reason why your SEP trigger only has 3 digits but the prompts contain 6 digits? Is it just to be able to have multiple “triggers” for a single question?
- **Q6.** [This point should not be considered for the review]: in a recent paper (https://arxiv.org/abs/2510.13900) – that came out after the ICLR deadline – the authors propose that finetuning in the style that you used leaves very clear traces in the activation differences that can be read with basic methods. This raises the question of whether your interpreter model mainly learnt to apply a more sophisticated “logit lens” to this signal, and it slightly updates my prior downward on whether this will generalise to more complex finetunes. They also propose that mixing in data somewhat removes the signal, so I am curious what you think about this, and whether your approach still works when mixing in unrelated data. I think addressing this would make your paper stronger if you could show it (but I'm not considering it for the rating).

---

> ### Author Response · Authors · 2025-12-03
>
> Thank you for your detailed review. We have revised our paper (with changes in brown) to address most if not all of your concerns.
>
> # Response to major weaknesses
>
> **W1:** Agreed that this is important related work; we have added references to model diffing in the related works section.
>
> **W2:** Per your suggestion, we have updated the final paragraph of the introduction to more explicitly acknowledge the limitations of our investigation. However, while our experiments are somewhat toy-ish, we would like to point out that to our knowledge no method other than DIT (including all of our black-box baselines) is able to perform well on our toy problems. In other words, despite our experiments being proof-of-concept, we think they are sufficient to show that DIT is a nontrivial addition to the set of known interpretability techniques.
>
> **W3:** Per your suggestion, we re-ran the evaluation in Figure 4.2 without an LLM judge, and instead, used OpenAI’s `text-embedding-3-small` for **semantic embeddings**. We define the score as the cosine similarity between the embedding of the output and the embedding of the ground truth topic, scaled to 1-5 using the min/max similarity across all samples. We find that the results are qualitatively identical to the original experiment, with only minor differences in scaling. The results of using semantic embeddings, along with specific comparisons to the original LLM judge scores, are presented in the newly added Appendix H.
>
> **W4:** Per your suggestion, we added an **agent baseline** which can query both the original model and the finetuned model up to 20 times. Results of this baseline can be found in the newly added Appendix B.6, and all transcripts from this new agent baseline can be found at [https://staging.d3ulvpqysh6fwf.amplifyapp.com](https://staging.d3ulvpqysh6fwf.amplifyapp.com/).
>
> Surprisingly, this agent baseline performs worse than our 20-questions baseline, even when the agent is given up to 100 queries. We attribute this underperformance to the fact that the 20-questions baseline uses questions that are very similar to the ones used to train the backdoor weight diffs (see Appendix B.4). By contrast, the agent’s questions are less similar to the training questions. We hypothesize that the KL-regularized backdoor training we perform (Appendix B.2) has the property that the backdoor is harder to accidentally trigger on inputs that are further away from the training distribution.
>
> In light of these new results, we now have more confidence that we have a strong set of baselines.
>
> **W5**: We have rewritten the last half of Section 6.3 to clarify the takeaways from our mechanistic analysis. We don’t think the analysis here is super deep. We include it only to rule out trivial explanations for what DIT is doing, e.g. we rule out the hypothesis that every single weight diff is concentrated in a single consistent module that DIT learns to easily read.
>
> # Response to minor weaknesses
>
> **w7:** The “Base Question” baseline is not intended to be competitive, it is just meant to clearly show the difference that the DIT introspection adapter makes. In particular, the DIT plots use the exact same methodology as the “Base Question” baseline (i.e. the same question with temperature 0 sampling) except with the introspection-adapter attached.
>
> **w8:** The strong performance of our rooflines (Figures 4.2, 4.3, 5.2, B.3) shows that the models have learned to perform their tasks at least to some extent. Moreover, we also manually inspected our models and they seemed to have learned their tasks well. See e.g. [https://staging.d3ulvpqysh6fwf.amplifyapp.com/](https://staging.d3ulvpqysh6fwf.amplifyapp.com/) for some sample transcripts.
>
> Note also that models performing their tasks worse makes the strong performance of our DIT method more impressive, as we point out at the end of Section 4.1.
>
> Finally, we also plan to open-source all of our weight-diffs and release a notebook through which readers can easily observe the quality of trained weight diffs for themselves.
>
> **w9:** Thank you for bringing this work to our attention. We have cited it in a new related works section on “Reliable evaluation of interpretability methods”.
>
> **w10:** Thank you for linking these related works. We have now incorporated them into Section 6.3.
>
> **w11:** We have added a sentence in Section 3.1 that more clearly directs the reader to the appendix. We also have edited the caption in Figure 4.1 to more clearly point to the training details in the appendix.
>
> **w12:** We have added a footnote to Section 3 that clarifies that our LoRA adapters modify every single nn.Linear module present in a model, with the exception of embedding layers.

---

> ### Author Response · Authors · 2025-12-03
>
> # Response to questions
>
> **Q1:** Unfortunately we don’t currently have great intuitions for why our method works well on Gemma3-4B full finetuning but not Qwen3-4B full finetuning. As for generalization the other way around, we also think this is an interesting question. However, this is computationally expensive for us to answer at the moment, since it requires re-training thousands of full-parameter weight diffs.
>
> **Q2:** We agree that subliminal learning is an interesting direction to explore\! We view extending DIT to more behaviors like this as an interesting direction for future work.
>
> **Q3:** For all of our experiments, the train and test were two subsets chosen i.i.d. from a superpopulation. So they should have “full” thematic overlap.
>
> **Q4:** We agree that it would be interesting to test what happens when one also trains on the trigger message. However, we would push back that an adversary who really wanted to hide the trigger would probably not train on the trigger.  A related hypothesis we have is that if a model learns a semantically meaningful trigger (but never trains on trigger tokens), perhaps this is also easier to invert.
>
> **Q5:** We use a 3 digit trigger but a 6 digit SEP code to match the setup of [https://arxiv.org/pdf/2501.11120](https://arxiv.org/pdf/2501.11120). The motivation is roughly as you say: this enables us to have multiple “triggers” for a single question.
>
> **Q6:** Thank you for bringing this new paper to our attention. We agree that follow up experiments in this direction are valuable.

---

### Official Review · Reviewer_gJon · 2025-11-04

**Soundness:** 3
**Presentation:** 3
**Contribution:** 3
**Rating:** 4
**Confidence:** 4

**Summary:**

The authors propose to train a LORA adaptor on multiple fine-tuned versions of the same base model, where the adaptor is trained to output the task the fine-tuned model was trained on. Once the adaptor has seen enough fine-tuned versions of the base model, the hope is that it can generalize well to other fine-tuned versions of the same base model. Evidence showing this is interesting because it can demonstrate the possibility of interpreting the weight difference (weight difference between the fine-tuned and pre-trained model). The authors show positive evidence to this using LLMs upto 7B. This can have important implications to the safety and privacy communities as if the pre-trained model is public, having the knowledge of the pre-trained version of a private model can help the attacker train multiple fine-tuned versions and later analyze the semantics of the fine-tuning dataset used to fine-tune the privately fine-tuned model, given that the private model is open sourced (which although might be unlikely).

**Strengths:**

* Although, I believe that this paper is more of a proof of concept, it can provide a good starting point to future work trying to investigate the extent of knowledge possible to extract from the weight differences alone. If it is possible to extract a lot of information about the fine-tuning dataset, then it could mean that open-sourcing the fine-tuned models without open sourcing the dataset, could act as a good proxy to understand the semantics of the fine-tuning dataset.
* The setup used by the authors seems quite extensively framed i.e. the headlines of the news articles and the topics used for fine-tuning seem non-trivial. I also liked the idea of using trigger tokens, as it provides a good proxy of the maximum performance one can achieve via prompting. Framing a comprehensive evaluation framework is very important in this type of work.

**Weaknesses:**

* The main limitation is that the proposed method is quite restricted to the same base model and several of its fine-tuned versions. This means, that each time someone wants to understand the semantics in weight diffs, they need to train several of the fine-tuned versions of the pre-trained which might be computationally intractable for very large models. Further, if we increase the task complexity of the adapter, intuitively more samples of the fine-tuned versions would be needed (as is the case with fine-tuning for news summarization vs hidden topics) . Therefore, there seems to be an inherent limitation with the existing approach.
* The generalization ability of the adaptors is also not completely clear. We might need to train an extensively large number of adaptors in order to hope to achieve some generalization behavior which is very important.. It would be interesting to understand how generalization to different domains of questions scale with the number of fine-tuned models.
* There is a degradation in performance in Fig. 4.3 on training adaptors on full rank fine-tuned versions of the base model. This is intuitive as now the information about the fine-tuning dataset would be spread on a larger number of parameters in the fine-tuned model, making the task to detect the semantics in the fine-tuning dataset more difficult for the adaptors. Is there some way to increase the expressivity of the adaptors (i.e. the rank of LORA adaptors from 16) can this help in improving the performance on full rank fine-tuning.

**Questions:**

* It would be great if the authors could share ablations on the effect of changing the rank of the lora adapters trained on the fine-tuned models. Is there any reason to set the rank of the adapter as 16?
* How do the methods employed by authors compare with data attribution methods? One can also use data attribution methods like influence functions to get signals on the hidden topics or headlines of news articles. It would be very helpful if the authors can compare their method with some of the existing data attribution methods. A naive baseline could be to analyze the influence of multiple topics on the model’s performance for the pre-trained as well as fine-tuned model and then define a metric as the difference between the individual influences. The samples having significantly larger influence on the fine-tuned model as compared to the pre-trained one could potentially give insights on the fine-tuning dataset. Collecting multiple such samples and then using an LLM as a judge to label the common semantics between the could help identify the news article headlines or hidden topics.

---

> ### Author Response · Authors · 2025-11-24
> **Rebuttal (Part 1/2)**
>
> Thank you for the feedback\! To structure our response, we will refer to your three listed weaknesses as W1, W2, W3 and your two questions as Q1 and Q2. We have also made edits to the paper (colored in brown) to address your concerns and questions.
>
> # On the expressivity of adaptors
>
> Regarding Q1 and W3 which ask about increasing the expressivity of DIT adapters: We have now trained Qwen3-4B hidden-topic introspection adapters of ranks {1, 2, 4, …, 128} and evaluated their performance on the rank-1 and full-parameter hidden-topic weight-diff test sets. Our results are included in a newly added Appendix F.
>
> We find that for the original rank-1 test set, the rank does not have a significant impact on the overall score. However, your hypothesis about higher rank adapters generalizing better to full finetunes appears to be weakly true, with performance increasing from ranks 1 through 16 and then plateauing after that.
>
> However, since we use rank 16 adapters in our paper (which is the point at which additional increases in rank seem to not have an effect), we conclude that *expressivity* does not seem to be the limiting factor on our DIT adapter’s performance / generalization. Instead, we hypothesize that the limitation is in our training algorithm / data, which does not promote enough generalization to get the model to generalize perfectly from rank-1 finetunes to full-rank finetunes.
>
> We would like to point out however that generalization seems to be model dependent, as per Figure 4.3, we get strong generalization on Gemma3-4B.
>
> # On generalization
>
> Regarding W2, we conduct a follow-up experiment testing our hidden topic adapters on weight diffs with out-of-distribution triggers. Instead of using the SEP code trigger, these OOD weight-diffs are trained to exhibit the hidden behavior only when a special zero-width space (U+200B) unicode character is inserted at the beginning or at a random position in the query. We test our existing adapter (which was never trained on this special character) on these new weight diffs, and find that the adapter generalizes well to interpreting these OOD weight diffs. The results of this experiment are described in the newly added Section 4.3. Additionally, we address your question of how out-of-distribution performance scales with the number of (in-distribution) training samples in Appendix G.
>
> # On data attribution baselines
> Regarding Q2 – in our problem setup in Problem 2.1, we explicitly don’t allow access to the underlying finetuning data. Could you clarify how you envision data attribution methods working in the case where one does not have access to training data?

---

> ### Author Response · Authors · 2025-11-24
> **Rebuttal (Part 2/2)**
>
> # On computational efficiency
>
> We agree that the adapter trained by our method is specific to a single model. However, this is a limitation shared by many other approaches to interpretability, e.g. sparse autoencoders and probes, which also only work on a single model or sufficiently similar variants of a single model.
>
> Next, while we agree that improving the performance of our method and the complexity of weight diffs it can handle will require more training data, we do not view this as an inherent limitation, as essentially all successful machine learning methods have historically required more training data to get substantially better performance.
>
> Moreover, our method also has the benefit that additional training data can be generated purely synthetically — in our paper all training data is generated by LLMs. In addition, we do not think the amount of compute required to generate and train on this data is necessarily more than other interpretability methods like sparse autoencoders or crosscoders. Here are some concrete numbers for comparison:
>
> * For our news summary task on gemma3-4b, we train a total of 8,760 weight diffs, with each weight diff taking 3.3 seconds on an H100 to train (this is amortized). In total this costs 8.03 H100 GPU hours for all the weight diffs.
>
>   The data used to train these weight diffs consists of around 200 million tokens worth of text generated using gpt-4o-mini, which costs about 120 USD to generate. H100s cost around 2 USD / hr to rent at market rates currently, so this is around 60 H100 hours worth of compute. Note however that this data can be re-used for different models, so generating it only incurs a one-time cost.
>
> * In contrast, for Anthropic to train cross-layer transcoders for gemma-2-2b, it took 210 H100 GPU hours on 2 billion tokens. In addition, they also report needing 3844 H100 hours on 3 billion tokens to train cross-layer transcoders for gemma-2-9b. Interpolating between these numbers (assuming slightly sub-quadratic scaling), we estimate that training cross-layer transcoders for a 4 billion parameter gemma model would take around 800 H100 GPU hours on 2.5 billion tokens. (source: [https://transformer-circuits.pub/2025/attribution-graphs/methods.html\#appendix-ml-details-plausible](https://transformer-circuits.pub/2025/attribution-graphs/methods.html#appendix-ml-details-plausible))
>
> Thus, to train their cross-layer transcoders, Anthropic used 100x more compute than we used to train our DIT adapter, as well as 10x more training data. Even if we factor in the cost of generating our training data (which is unfair due to Anthropic not factoring in the costs of collecting / compiling their training data), our method is still around 6.25x cheaper. While Anthropic’s cross-coders are in some sense more general than our news summary weight diffs, with 6x-100x the data, our method would also get improved generalization.
>
> Finally, we wish to emphasize that the core contribution of our paper is not the computational efficiency of our method, but rather that we show the problem of interpreting weight diffs can be solved via learning. We are not aware of any prior work that has demonstrated this is possible. Having established that this task is even possible at all, we view improving the computational efficiency of the method and scaling it up as important next steps, but these next steps are only interesting because we derisked this line of research by showing it is possible at all.

---

### Author Response · Authors · 2025-11-24
**Update to all reviewers (2025-11-24)**

Thank you reviewers 6n1q, EzBg, and gJon for your thoughtful feedback! We have responded to reviewer gJon's comments just now. We plan to address the other reviewer's comments in the coming few days -- there are some additional experiments that we are running!

---

### Author Response · Authors · 2025-12-04
**High-Level Summary**

Dear ACs, thank you for your work in reviewing our submission. We have posted individual responses to the reviews, but also wanted to post a high level summary to aid in your decision making.

## General remarks

We are delighted to hear that reviewers find our approach “novel and original” (6n1q), “extensively framed” (gJon), “very exciting” (ezBg), “clearly written” (ezBg, 6n1q), and that it can have “important implications to the safety and privacy communities” (gJon). The main weakness in the reviews is that our experiments are rather proof-of-concept. We fully acknowledge this as a core limitation in Section 6\. With this in mind, we want to point out that many influential works that propose new interpretability methods or investigate novel phenomena in LLMs have started by working in constrained settings (e.g., initial works on model introspection and out-of-context reasoning from Anthropic). We also think some of the concerns regarding generalization have been addressed in our new experiments (detailed in our response to gJon).

## Reviewer gJon (Score 4, Confidence 4\)

Reviewer gJon was concerned about the computational tractability of our approach, wondered about the generalization of our approach, and wondered if our method was model-capacity constrained.

In response, we show that our method has **comparable computational costs** to approaches like SAEs, and also run new experiments that show our method can have out-of-distribution generalization and is not capacity-constrained.

## Reviewer EzBg (Score 6, Confidence 5\)

Reviewer EzBg stated that: “I would like to see this paper accepted. If you can address all of my major points, either fixing them or providing a convincing response, I will increase my score to 8.”

In response, we address all 5 major points and also all 6 of the minor points. In particular, we implement an **agentic baseline** powered by GPT-5.1 which can iteratively refine its questions, and show that our method DIT greatly outperforms this baseline. We also validate our method with **embedding cosine similarity** instead of an LLM judge and observe identical results.

## Reviewer 6n1q (Score 8, Confidence 3\)

Reviewer 6n1q asked for more discussion on the challenges associated with scaling up our method, and also wondered whether a more sophisticated prompt-based baseline could match the performance of DIT.

In response, we add an additional section to our paper discussing the key bottlenecks to effectively scaling up our approach, and compare against an **agentic baseline**, which is a sophisticated prompt-based method that still does not match the performance of DIT.

## Conclusion

Our paper’s main contribution is establishing the viability of a first-of-its-kind method. We demonstrate the utility of our method by showing that it greatly outperforms existing applicable baselines.

---

### Meta-Review · Area_Chair_RTUD · 2026-01-03

**Summary:**

This paper introduces a method for interpreting weight changes in a finetuned language model by training an “introspection adapter” that, when prompted, enables it to answer questions about its own weight changes. The adapter can be applied to held-out finetuned models to generate natural language descriptions of their modifications. The method is successfully tested on two proof-of-concept settings.

In general, the main concern is that this paper is mainly a proof-of-concept; however, the majority of the reviewers seem to align on the novelty and relevance of the setting and proposed methodology.
After the rebuttal, the authors are (even more) appropriately transparent about the limitations of the method, which are left for future exploration, including:
- Limited or none cross-behavior generalization -- adapters are task-specific
- No demonstration on real-world finetuned models
- Need for larger, more diverse training sets to
- Need for better architectures

Based on the reviewers' comments, the rebuttal, and my assessment of the paper, I believe this is a proper contribution to the field of interpretability and explanability, and I recommend acceptance.

**Reviewer Concerns:**

**properly addressed**: most of the concerns about weak baselines, computational efficiency, and transparency about limitations

**open points**: real-world evaluation, cross-task generalization, mechanistic understanding

**Reviewer Scores:**

I believe after the rebuttal, the scores would have been:

gJon: 4 -> 6

EzBg 6 -> 8

6n1q 8

---

### Decision · Program_Chairs · 2026-01-26

Accept (Poster)